# Age at diagnosis, glycemic trajectories, and responses to oral glucose-lowering drugs in type 2 diabetes in Hong Kong: A population-based observational study

**Calvin Ke**[1,2], **Thérèse A. Stukel**[3,4], **Baiju R. Shah**[2,3,4,5], **Eric Lau**[1,6], **Ronald C. Ma**[1,7], **Wing-Yee So**[1], **Alice P. Kong**[1,7], **Elaine Chow**[1], **Juliana C. N. Chan**[1,6,7]*, **Andrea Luk**[1,6,7]

1 Department of Medicine and Therapeutics, The Chinese University of Hong Kong, Prince of Wales Hospital, Shatin, Hong Kong SAR, China, 2 Department of Medicine, University of Toronto, Canada, 3 Institute of Health Policy, Management and Evaluation, University of Toronto, Canada, 4 ICES, Toronto, Canada, 5 Department of Medicine, Sunnybrook Health Sciences Centre, Toronto, Canada, 6 Asia Diabetes Foundation, Metropole Square, Shatin, Hong Kong SAR, China, 7 Hong Kong Institute of Diabetes and Obesity and Li Ka Shing Institute of Health Science, The Chinese University of Hong Kong, Prince of Wales Hospital, Shatin, Hong Kong SAR, China

* jchan@cuhk.edu.hk

**Data Availability Statement:** Due to local laws and regulations regarding the use and distribution of personal data, the data used in the present study

## Abstract

### Background

Lifetime glycemic exposure and its relationship with age at diagnosis in type 2 diabetes (T2D) are unknown. Pharmacologic glycemic management strategies for young-onset T2D (age at diagnosis <40 years) are poorly defined. We studied how age at diagnosis affects glycemic exposure, glycemic deterioration, and responses to oral glucose-lowering drugs (OGLDs).

### Methods and findings

In a population-based cohort ($n$ = 328,199; 47.2% women; mean age 34.6 and 59.3 years, respectively, for young-onset and usual-onset [age at diagnosis ≥40 years] T2D; 2002–2016), we used linear mixed-effects models to estimate the association between age at diagnosis and A1C slope (glycemic deterioration) and tested for an interaction between age at diagnosis and responses to various combinations of OGLDs during the first decade after diagnosis. In a register-based cohort ($n$ = 21,016; 47.1% women; mean age 43.8 and 58.9 years, respectively, for young- and usual-onset T2D; 2000–2015), we estimated the glycemic exposure from diagnosis until age 75 years.

People with young-onset T2D had a higher mean A1C (8.0% [standard deviation 0.15%]) versus usual-onset T2D (7.6% [0.03%]) throughout the life span ($p$ < 0.001). The cumulative glycemic exposure was >3 times higher for young-onset versus usual-onset T2D (41.0 [95% confidence interval 39.1–42.8] versus 12.1 [11.8–12.3] A1C-years [1 A1C-year = 1 year with 8% average A1C]). Younger age at diagnosis was associated with faster glycemic deterioration (A1C slope over time +0.08% [0.078–0.084%] per year for age at diagnosis 20 years

cannot be deposited in a public repository. Data access can be applied for through the Data Sharing Portal of the Hong Kong Hospital Authority (https://www3.ha.org.hk/data/DCL/Index/).

**Funding:** CK is supported by the Canadian Institutes of Health Research (https://cihr-irsc.gc.ca/) Canada Graduate Scholarship and Michael Smith Foreign Study Supplements, the University of Toronto Clinician Investigator Program (https://cip.utoronto.ca/), the Canadian Society of Endocrinology and Metabolism (https://www.endo-metab.ca/) Dr. Fernand Labrie Research Fellowship Grant, and the Royal College of Physicians and Surgeons of Canada (http://www.royalcollege.ca/) Detweiler Traveling Fellowship. (Grant numbers not applicable.) The funders had no role in study design, data collection and analysis, decision to publish, or preparation of the manuscript.

**Competing interests:** I have read the journal's policy and the authors of this manuscript have the following competing interests: RCWM acknowledges receiving research support (outside of this work) from AstraZeneca, Bayer, and Pfizer for conducting clinical trials and honoraria or consultancy fees from AstraZeneca and Boehringer Ingelheim, all of which has been donated to the Chinese University of Hong Kong to support diabetes research. RCWM is a member of the Editorial Board of PLOS Medicine. AOYL acknowledges receiving research support (outside of this work) from Boehringer Ingelheim, MSD, Sanofi, and Amgen and travel grants from travel grant from AstraZeneca, Boehringer Ingelheim, MSD, Novartis, Novo Nordisk, and Sanofi. JCNC and RCWM are cofounders of GemVCare, a diabetes genetic testing laboratory, which was established through support from the Technology Start-up Support Scheme for Universities (TSSSU) from the Hong Kong Government Innovation and Technology Commission (ITC). JCNC is the Chief Executive Officer, on a pro bono basis, of the Asia Diabetes Foundation (ADF), which is a nonprofit research organization which designed and implemented the Joint Asia Diabetes Evaluation (JADE) Technology as an extension to the HKDR, under the governance of the CUHK Foundation. The HKDR was established as a research-driven quality improvement program initiated by the Chinese University of Hong Kong (CUHK)-Prince of Wales Hospital Diabetes Care and Research Team, supported by the Hong Kong Foundation for Research and Development in Diabetes established at CUHK. In 2007, this was merged with the web-based JADE Technology, complete with care protocols, risk stratification, personalized reporting, and decision support. The JADE Technology was

versus +0.02% [0.016–0.018%] per year for age at diagnosis 50 years; *p*-value for interaction <0.001). Age at diagnosis ≥60 years was associated with glycemic improvement (−0.004% [−0.005 to −0.004%] and −0.02% [−0.027 to −0.0244%] per year for ages 60 and 70 years at diagnosis, respectively; *p*-value for interaction <0.001). Responses to OGLDs differed by age at diagnosis (*p*-value for interaction <0.001). Those with young-onset T2D had smaller A1C decrements for metformin-based combinations versus usual-onset T2D (metformin alone: young-onset −0.15% [−0.105 to −0.080%], usual-onset −0.17% [−0.179 to −0.169%]; metformin, sulfonylurea, and dipeptidyl peptidase-4 inhibitor: young-onset −0.44% [−0.476 to −0.405%], usual-onset −0.48% [−0.498 to −0.459%]; metformin and α-glucosidase inhibitor: young-onset −0.40% [−0.660 to −0.144%], usual-onset −0.25% [−0.420 to −0.077%]) but greater responses to other combinations containing sulfonylureas (sulfonylurea alone: young-onset −0.08% [−0.099 to −0.065%], usual-onset +0.06% [+0.059 to +0.072%]; sulfonylurea and α-glucosidase inhibitor: young-onset −0.10% [−0.266 to 0.064%], usual-onset: 0.25% [+0.196% to +0.312%]). Limitations include possible residual confounding and unknown generalizability outside Hong Kong.

## Conclusions

In this study, we observed excess glycemic exposure and rapid glycemic deterioration in young-onset T2D, indicating that improved treatment strategies are needed in this setting. The differential responses to OGLDs between young- and usual-onset T2D suggest that better disease classification could guide personalized therapy.

## Author summary

### Why was this study done?

- Young-onset type 2 diabetes (diagnosed before age 40 years) is an aggressive disease, associated with higher risks of mortality and other complications compared with usual-onset type 2 diabetes (diagnosed at age 40 years or after).

- Although exposure to hyperglycemia is a key risk factor for type 2 diabetes complications, the magnitude of this exposure over a lifetime has never been quantified.

- Because young people were excluded from randomized control trials of oral glucose-lowering drugs, it is unknown whether these drugs are effective for treating young-onset type 2 diabetes.

### What did the researchers do and find?

- Young-onset type 2 diabetes was associated with over triple the exposure to hyperglycemia from diagnosis until age 75 years compared with usual-onset type 2 diabetes.

- Diabetes progressed much faster in young-onset type 2 diabetes compared with usual-onset type 2 diabetes.

designed and implemented by the ADF to enable other clinics and hospitals to establish diabetes registers and contribute anonymized data for research purposes. The ADF was set up as a charitable research organization governed by the CUHK Foundation.

**Abbreviations:** A1C, hemoglobin $A_{1c}$; MACE, major adverse cardiovascular events; OGLD, oral glucose-lowering drug; T2D, type 2 diabetes.

- Compared with usual-onset type 2 diabetes, people with young-onset type 2 diabetes had smaller responses to most metformin-based drug combinations and greater responses to other combinations containing sulfonylureas.

## What do these findings mean?

- Excess exposure to hyperglycemia and rapid disease progression in young-onset T2D call for better treatment strategies.

- The differential responses to oral glucose-lowering drugs between young- and usual-onset T2D suggest that better disease classification could guide personalized therapy.

## Introduction

Life-threatening complications of type 2 diabetes (T2D) are caused by long-term exposure to hyperglycemia [1]. Glycemic exposure is defined as the area under the glycated hemoglobin $A_{1c}$ (A1C) curve in excess of 7% over time [2–5]. A meta-analysis of randomized control trials showed that every 10 A1C-years of glycemic exposure (that is, 10 years of A1C at 8%) predicts a 25% increase in the relative risk of major adverse cardiovascular events (MACE) [2]. Young-onset T2D (defined here as age at diagnosis <40 years) is an aggressive phenotype associated with higher lifetime risks of MACE and other complications compared to usual-onset T2D (age at diagnosis ≥40 years) [6–9]. Although early age at diagnosis and poor glycemic control are important risk factors for complications [10], lifetime glycemic exposure and its relationship with age at T2D diagnosis are unknown.

Reducing glycemic exposure typically requires frequent escalation of glucose-lowering therapies to overcome the natural history of T2D, which is characterized by progressively declining β-cell function and insulin sensitivity [11]. The rate of β-cell function decline is known as glycemic deterioration [3,12,13]. This measure can be quantified using repeated measurements of A1C over time, after accounting for glucose-lowering drugs using statistical adjustment [13]. Glycemic deterioration is more precise than other measures using binary thresholds (e.g., treatment failure) [14]. Among Europeans with usual-onset T2D, younger age at diagnosis is associated with faster glycemic deterioration [13,14]. In Asia, 1 in 5 adults with T2D attending medical clinics has young-onset T2D [10], yet data on glycemic exposure and deterioration are lacking.

Similarly, pharmacologic glycemic management strategies for young-onset T2D are poorly defined because this population was excluded from trials of glucose-lowering therapies in adults [15]. In a pediatric trial, metformin alone was associated with treatment failure in 51.7% of children with T2D [16]. Another study of children with T2D found that 3 months of insulin and 9 months of metformin failed to improve β-cell function versus metformin alone [17]. Considering the lack of randomized trials among adults with young-onset T2D [15], observational studies may provide valuable insights in this understudied population.

To address these knowledge gaps, we conducted a large register- and population-based cohort study to measure how age at diagnosis affects (1) glycemic exposure, (2) glycemic deterioration, and (3) responses to oral glucose-lowering drugs (OGLDs) during the first decade after diagnosis among adults with T2D. We hypothesized that earlier age at diagnosis would be

associated with increased glycemic exposure, faster glycemic deterioration, and decreased responsiveness to OGLDs.

## Methods

### Setting

Hong Kong has a population of 7.3 million people, 92% of whom are of Chinese ethnicity [18]. The estimated diabetes prevalence was 10.3% in 2014 [19]. The Hong Kong Hospital Authority (HA) provides universal public healthcare modeled after the British National Health Service. Because of the high out-of-pocket cost of private healthcare, 95% of people with diabetes in Hong Kong receive care in HA clinics [20]. Consultation, prescription, and medication dispensing services are all provided on site under an all-inclusive nominal user fee [21], which is waived for low-income and other vulnerable groups [22]. All hospitals and clinics managed by the HA share the same electronic health record (EHR) with data including laboratory tests, discharge summaries, and dispensed prescriptions. Dispensed prescription records are comprehensive because drugs are prescribed and dispensed on site at the time of consultation. These data are linked by the unique Hong Kong Identity Card number.

### Data sources

In 1995, the Prince of Wales Hospital set up the multicenter Hong Kong Diabetes Register (HKDR) as a research-driven quality improvement program [23]. The HKDR is a prospective cohort of people with prevalent diabetes, of varying disease duration. Participants were enrolled from 1994 to 2015. All participants undergo structured assessment (eyes, feet, blood, urine) by trained nurses every 2–3 years to collect data not routinely captured in the HA EHR, including diabetes type, age at diagnosis, family history, and lifestyle habits. Enrollment was open to any adult with diabetes based on self- or physician-initiated referrals from community- and hospital-based clinics. All services were provided at the nominal fee charged by the HA.

The Hong Kong Diabetes Surveillance Database (HKDSD) is a population-based cohort of incident diabetes cases from across the entire territory, identified from the HA EHR. We defined diabetes as any person with an A1C ≥6.5% (48 mmol/mol), outpatient fasting plasma glucose ≥7 mmol/L, non-insulin glucose-lowering drug prescription, or long-term insulin prescription (≥28 days). The date of diagnosis is defined as the date of first occurrence of any of these events. According to HA regulations, these drugs are only indicated for diabetes [24]. To avoid detecting gestational diabetes, events occurring within 9 months of any pregnancy-related encounter are excluded. As diabetes type is not systematically encoded in this EHR, we previously developed and validated algorithms to classify diabetes type based on diagnosis codes and prescriptions [25]. In this study, we excluded people who received their first insulin prescription within 90 days of diabetes diagnosis as having type 1 diabetes [25] and classified the remainder as T2D. This definition has a sensitivity of 94.6% (95% confidence interval 93.9%–95.2%) and positive predictive value of 99.9% (99.8%–100.0%) for predicting T2D [25].

To estimate lifetime glycemic exposure, we required person-years of follow-up from across the life span. This requirement necessitated the inclusion of people with new (incident) and old (prevalent) diagnoses of T2D of varying disease duration. We used the HKDR ("Register cohort") for this objective because it includes people with both new and old T2D cases with disease duration ranging from zero to more than 70 years, whereas the HKDSD only includes new T2D cases after 2002. For example, a 70-year-old person in 2010 who was diagnosed with T2D at age 30 years in 1970 would be included in the HKDR but not the HKDSD. To measure glycemic deterioration (defined as A1C slope over time, after statistical adjustment for

OGLDs) and medication responses, we required person-years of follow-up from the first decade after a new T2D diagnosis. This requirement was based on a previous study showing that glycemic deterioration is a linear function of time during this period [26]. Although both cohorts included people with new diagnoses, we used the HKDSD ("Population cohort") because it was larger and included people in the HKDR.

## Study population

**Register cohort (glycemic exposure).** We included adults aged 18–75 years in the HKDR with prevalent T2D observed between January 1, 2000, and December 31, 2015, and defined the index date as the earliest date within this period when the person met these criteria (Fig A in S1 Appendix). We excluded observations between 1994 to 1999 to more closely match the time period of the population cohort. We excluded people with type 1 diabetes, gestational diabetes, and non-Chinese ethnicity.

**Population cohort (glycemic deterioration and medication responses).** We included adults aged 18–75 years in the HKDSD with T2D diagnosed between January 1, 2002, and December 31, 2012, (Fig A in S1 Appendix; dates chosen based on data availability). The index date was the diagnosis date. We followed people for up to 10 years after diagnosis of diabetes, censoring at the date of the first insulin prescription. The maximum follow-up date was December 31, 2016, for both the register and population cohorts, censoring at death or age 75 years.

## Statistical analysis

We conducted the study according to a prospective analysis plan (S1 Analysis Plan). The primary outcome was A1C measured over the follow-up period, and the primary exposure was age at diagnosis, expressed as a continuous variable. In the register cohort, we calculated the mean A1C by attained age among all people with young-onset T2D. Using these values, we estimated the glycemic exposure, defined as the area under the A1C curve in excess of 7% (53 mmol/mol), from the mean observed age at diagnosis until age 75 years (Fig B in S1 Appendix), and repeated this procedure for usual-onset T2D.

In the population cohort, we used a linear mixed-effects model with age at diagnosis and time as independent variables and A1C as the dependent variable, with person-specific random intercepts and slopes because we assumed each person had his or her own A1C trajectory. Although we did not dichotomize age at diagnosis in the model, we presented the expected results for young- and usual-onset T2D based on the mean observed ages at diagnosis in each group. Glycemic deterioration was defined as the A1C slope over time, adjusted for model covariates. We included time-varying covariates for each OGLD based on their dispensing records, allowing for people to switch on and off a drug. We assumed the same absolute effect for each drug class, regardless of A1C level, and that drugs within the same class would lower A1C by a similar decrement [27]. We included metformin, sulfonylureas, and 10 multidrug combinations as unique variables (Table A in S1 Appendix). Effects of individual drugs taken in combination were not assumed to be additive (S1 Appendix) because OGLDs may have different effects when prescribed alone versus in combination [28,29]. Each drug combination was considered as a separate variable. When a new drug was dispensed, we excluded the first A1C measurement on the new treatment regimen to allow sufficient time for the A1C to equilibrate. We adjusted for prespecified variables (Fig C in S1 Appendix) including sex and recent comorbidities, namely, ischemic heart disease, congestive heart failure, stroke, peripheral arterial disease, and cancer, based on principal diagnoses from hospitalizations occurring within 2 years prior to the index date (Table B in S1 Appendix) and chronic kidney disease classified by the

estimated glomerular filtration rate. As anthropometric data were unavailable, we included triglycerides and high-density lipoprotein cholesterol as proxies for obesity [30].

We estimated the association between age at diagnosis and A1C slope (glycemic deterioration) and tested for an interaction between age at diagnosis and responses to various combinations of OGLDs. We conducted several sensitivity analyses. To test the validity of our assumption of a linear relationship between age at diagnosis and A1C, we repeated the analysis using a nonlinear model with restricted cubic splines containing 4 knots placed at fixed quantiles [31]. We also repeated the analysis excluding A1C values from the first 6 and 12 months after the diagnosis of T2D because these measurements might be unusually elevated as a consequence of transiently depressed β-cell function ("glucotoxicity") [32]. To test whether the baseline A1C level affected A1C lowering, we repeated the analysis (post hoc) with an interaction term between each drug combination and its observed baseline A1C level. Interaction terms between age at diagnosis and drug combinations were excluded in this model because of computational limitations.

We used the MIXED procedure (SAS version 9.4, SAS Institute, Cary, NC, www.sas.com), specifying the spatial power covariance structure to account for nonequal time intervals between A1C measurements. Missing outcome data were minimal (4.0% and 6.4% in the register and population cohorts) and handled by complete case analysis. The study was approved by The Chinese University of Hong Kong-New Territories East Cluster Clinical Research Ethics Committee and the University of Toronto Health Sciences Research Ethics Board. Data in the HKDSD were anonymized at the time of access. Individuals in the HKDR provided written informed consent for the use of their data. This study is reported as per the Strengthening the Reporting of Observational Studies in Epidemiology guideline (S1 STROBE checklist).

## Results

We included 21,016 people (47.1% women, 0.2 million person-years follow-up, median 8.4 years) in the register cohort and 328,199 people (47.2% women, 2.4 million person-years follow-up, median 7.9 years) in the population cohort with >3 million A1C measurements combined (Fig D in S1 Appendix). The mean age at diagnosis was similar in both cohorts (young-onset: 33.8–34.6 years, usual-onset: 53.8–59.3 years, Table 1). Recent comorbidities were more common in the register than the population cohort. The baseline A1C was higher in young-versus usual-onset T2D (register: young-onset, mean 7.8% [62 mmol/mol, standard deviation 1.5%]; usual-onset, 7.3% [56 mmol/mol, 1.2%]; population: young-onset, 7.6% [60 mmol/mol, 1.6%]; usual-onset, 7.3% [56 mmol/mol, 1.3%]). Lipid levels were similar across cohorts. Metformin (young-onset: 31.8–56.4%, usual-onset: 41.5–48.6%) and sulfonylureas (young-onset: 25.9–38.4%, usual-onset: 33.6–40.7%) were the most common OGLDs. Insulin use during the first year after diagnosis was more common in the register (young-onset: 24.2%, usual-onset: 11.7%) than the population cohort (young-onset: 3.0%, usual-onset: 1.1%).

### Glycemic exposure in young-onset and usual-onset T2D

In the register, people with young-onset T2D had a higher mean observed A1C (mean of annual means 8.0% [64 mmol/mol], standard deviation 0.15%) versus those with usual-onset T2D (7.6% [59 mmol/mol], 0.03%) throughout the life span ($p < 0.001$; Fig 1). The cumulative glycemic exposure was >3 times higher for young- versus usual-onset T2D (41.0 [39.1–42.8] versus 12.1 [11.8–12.3] A1C-years [each A1C-year equivalent to one year with an 8% (64 mmol/mol) average A1C]). In the population cohort, mean observed A1C levels were more elevated among people with young-onset compared with usual-onset T2D, both at diagnosis and throughout the first decade after diagnosis.

**Table 1. Baseline characteristics in the register (2000–2016) and population (2002–2016) cohorts.**

| | Register Cohort | | Population Cohort | |
| --- | --- | --- | --- | --- |
| | Young-Onset (*n* = 4,058) | Usual-Onset (*n* = 16,958) | Young-Onset (*n* = 15,265) | Usual-Onset (*n* = 312,934) |
| Age at diagnosis (years) | 33.8 (5.5) | 53.8 (8.1) | 34.6 (4.8) | 59.3 (9.0) |
| Index age (years) | 43.8 (10.5) | 58.9 (8.3) | 34.6 (4.8) | 59.3 (9.0) |
| Women (*n*, %) | 2,031 (50.0) | 7,876 (46.4) | 6,508 (42.6) | 148,516 (47.4) |
| Recent comorbidities within 2 year of index* (*n*, %) | | | | |
| Ischemic heart disease | 62 (1.5) | 546 (3.2) | 57 (0.4) | 4,780 (1.5) |
| Congestive heart failure | 14 (0.3) | 130 (0.8) | 28 (0.2) | 1,244 (0.4) |
| Stroke | 57 (1.4) | 534 (3.2) | 55 (0.4) | 3,809 (1.2) |
| Peripheral arterial disease | 11 (0.3) | 33 (0.2) | 2 (0.0) | 105 (0.0) |
| Cancer | 56 (1.4) | 359 (2.1) | 152 (1.0) | 4,794 (1.5) |
| Laboratory Values (within 2 years of index* for A1C; 3 years for other variables) | | | | |
| Hemoglobin A1C (%) | 7.8 (1.5) | 7.3 (1.2) | 7.6 (1.6) | 7.3 (1.3) |
| Fasting plasma glucose (mmol/L) | 8.6 (2.4) | 7.9 (1.9) | 8.1 (2.6) | 7.6 (2.0) |
| LDL-C (mmol/L) | 2.8 (0.7) | 2.7 (0.7) | 2.9 (0.8) | 3.0 (0.8) |
| HDL-C (mmol/L) | 1.3 (0.3) | 1.3 (0.3) | 1.2 (0.3) | 1.1 (0.3) |
| Triglycerides (mmol/L; median, IQR) | 1.5 (1.2) | 1.5 (1.0) | 1.7 (1.4) | 1.5 (1.0) |
| Estimated GFR (mL/min/1.73 m$^2$) | 93.2 (22.7) | 78.9 (20.5) | 105 (18.0) | 81.2 (18.9) |
| <60 mL/min/1.73 m$^2$ (*n*, %) | 327 (8.0) | 2,787 (16.4) | 404 (2.8) | 36,541 (12.6) |
| <15 mL/min/1.73 m$^2$ (*n*, %) | 41 (1.0) | 163 (1.0) | 59 (0.4) | 2,017 (0.7) |
| Pharmacotherapy within 1 year after index* (*n*, %) | | | | |
| Metformin | 1,292 (31.8) | 7,032 (41.5) | 8,611 (56.4) | 152,156 (48.6) |
| Sulfonylurea | 1,051 (25.9) | 5,704 (33.6) | 5,856 (38.4) | 127,512 (40.7) |
| DPP-4 Inhibitor | 16 (0.4) | 68 (0.4) | 417 (0.1) | 43 (0.3) |
| Thiazolidinedione | 38 (0.9) | 124 (0.7) | 178 (0.1) | 13 (0.1) |
| Acarbose | 24 (0.6) | 96 (0.6) | 42 (0.3) | 1,085 (0.4) |
| GLP-1 receptor agonist | 0 (0.0) | 0 (0.0) | 3 (0.0) | 2 (0.0) |
| Insulin | 983 (24.2) | 1,983 (11.7) | 464 (3.0) | 3,400 (1.1) |

Young-onset type 2 diabetes is defined here as age at diagnosis <40 years and usual-onset type 2 diabetes as ≥40 years. Values are means and standard deviations unless otherwise indicated. Because of the large sample size, baseline differences should be interpreted based on clinical significance rather than statistical significance.

*The index date in the register cohort was the date of enrollment in the register, whereas the index date in the population cohort was the date of diabetes diagnosis.

DPP-4, dipeptidyl peptidase-4; GFR, glomerular filtration rate; GLP-1, glucagon-like peptide 1; HDL-C, high-density lipoprotein cholesterol; IQR, interquartile range; LDL-C, low-density lipoprotein cholesterol.

## Glycemic deterioration in young-onset and usual-onset T2D

In the population cohort, glycemic deterioration differed significantly across age at diagnosis (Fig 2, Table C in S1 Appendix). Younger age at diagnosis was associated with faster glycemic deterioration (*p*-value for interaction <0.001). People aged ≤30 years at diagnosis had the most rapid deterioration (+0.08% [95% confidence interval 0.078 to 0.084%] per year for age 20 years at diagnosis) as compared with no deterioration in those aged 60 years at diagnosis (0.00% [−0.005 to −0.004%] per year), and glycemic improvement (−0.02% [−0.027 to −0.0244%] per year) in people aged 70 years at diagnosis. In a sensitivity analysis, we observed similar results allowing for a nonlinear relationship between age at diagnosis and glycemic deterioration (Fig E in S1 Appendix) and excluding A1C values from the first 6 to 12 months after diagnosis (Fig F in S1 Appendix).

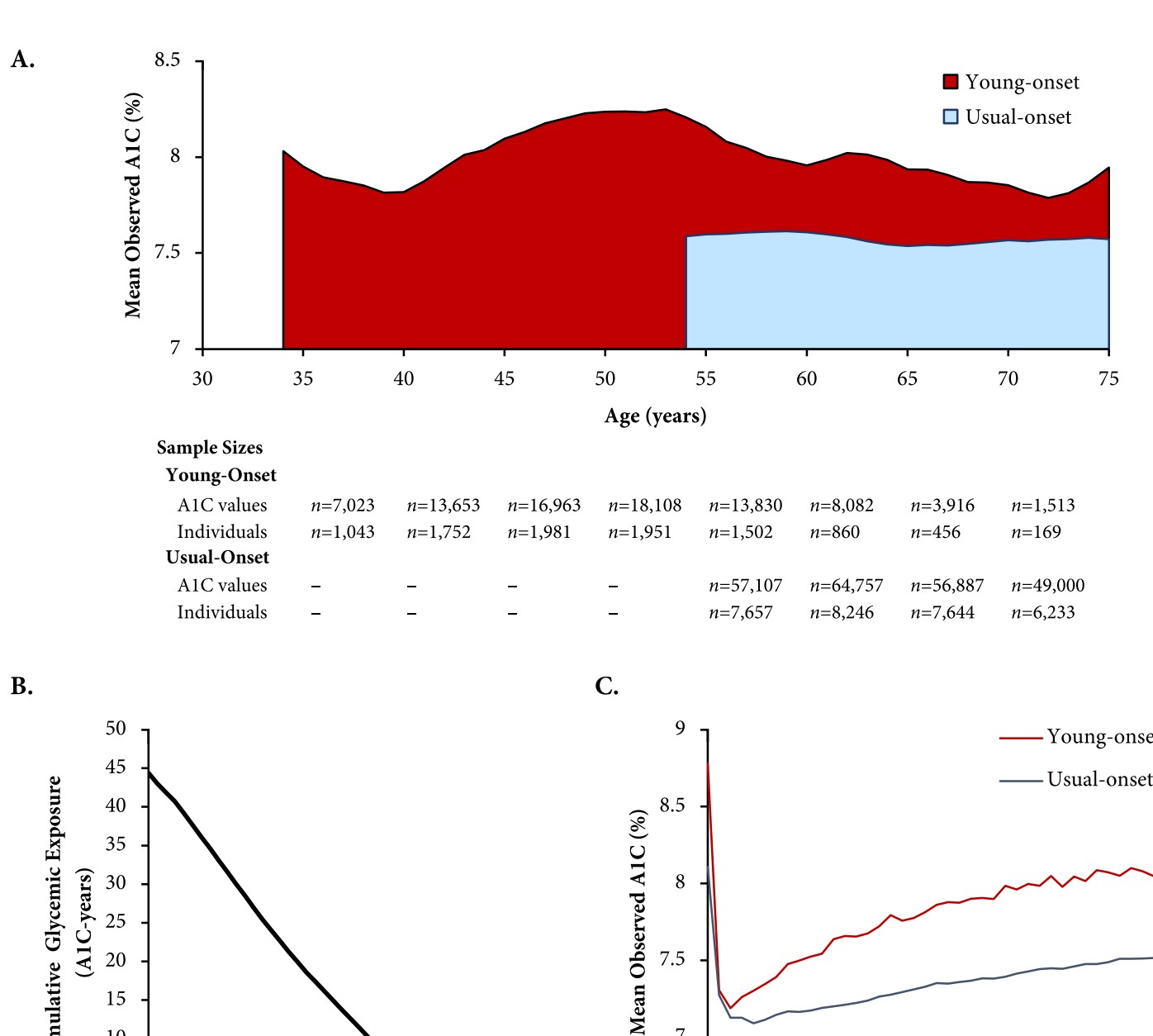

**Fig 1. Observed A1C and glycemic exposure among adults in the register and population cohorts.** These A1C values are not adjusted for medications. (A) Mean observed A1C across the age span (attained age) in the register cohort (2000–2016), stratified by age at diagnosis (smoothed using 3-year moving averages). The shaded areas indicate glycemic exposure in excess of 7%. Sample sizes are indicated for each 5-year age group. (B) Cumulative glycemic exposure in the register cohort (2000–2016), defined as area under the A1C curve in excess of 7%, by age at diagnosis. One A1C-year is equivalent to one year of exposure to an average A1C of 8%. For example, a person diagnosed with diabetes at age 30 years, with an average A1C of 8% from age 30 to 75 years, would have been exposed to a 1% excess in A1C over 45 years, which is equivalent to 45 A1C-years of glycemic exposure. (C) Mean observed A1C by years since type 2 diabetes diagnosis in the population cohort (2002–2016), stratified by age at diagnosis. A1C, hemoglobin $A_{1c}$.

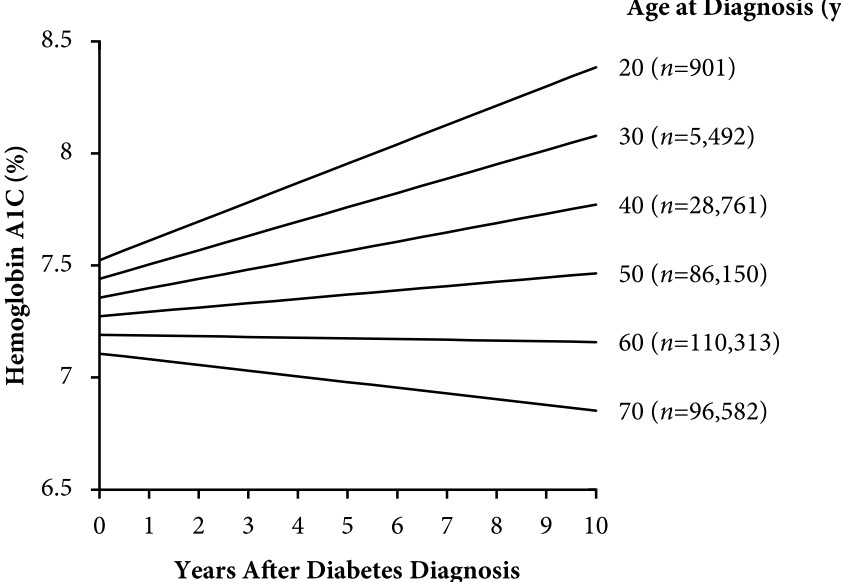

**Fig 2. Glycemic deterioration during the first decade after type 2 diabetes diagnosis.** Results are stratified by age at diagnosis (population cohort, Hong Kong Diabetes Surveillance Database, 2002–2016). Glycemic deterioration is the modeled slope of the A1C over time after adjusting for oral glucose-lowering drug prescriptions. The sample size ($n$) is indicated for each age group (age at diagnosis <25, 25–34, 35–44, 45–54, 55–64, ≥65 years). See Table C in S1 Appendix for numeric values. A1C, hemoglobin A1c.

## Responses to OGLDs in young-onset and usual-onset T2D

There was a statistically significant difference between people with young- and usual-onset T2D in their responses to OGLDs ($p$-value <0.001 for omnibus test across all combinations; Fig 3). Combinations containing metformin were associated with lower A1C values, but these

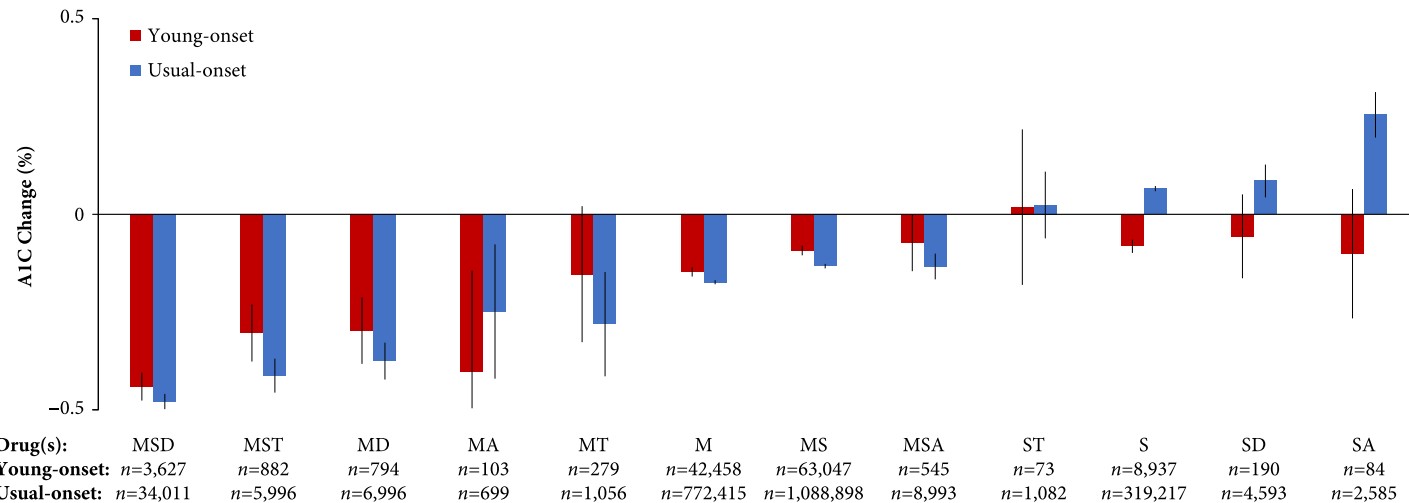

**Fig 3. A1C responses to oral glucose-lowering drugs among people with young- and usual-onset type 2 diabetes.** These insulin-naive individuals were observed during the first decade after diabetes diagnosis (population cohort, Hong Kong Diabetes Surveillance Database, 2002–2016). Sample sizes (number of A1C measurements) are indicated for each combination. Error bars indicate 95% confidence intervals. Differences between young- and usual-onset type 2 diabetes were statistically significant across all combinations (omnibus test $p < 0.001$). A, acarbose; D, dipeptidyl peptidase-4 inhibitor; M, metformin; S, sulfonylurea; T, thiazolidinedione.

decrements appeared slightly smaller for young-onset T2D. Metformin alone was associated with an A1C decrement of −0.15% (95% confidence interval −0.105% to −0.080%) in young-onset and −0.17% (−0.179% to −0.169%) in usual-onset T2D. The combination of metformin, a sulfonylurea and a dipeptidyl peptidase-4 inhibitor had the greatest A1C decrement (young-onset: −0.44% [−0.476% to −0.405%], usual-onset: −0.48% [−0.498% to −0.459%]). The combination of metformin and an α-glucosidase inhibitor (class exclusively consisting of acarbose) was associated with a greater A1C decrement in young-onset (−0.40% [−0.660% to −0.144%]) than usual-onset T2D (−0.25% [−0.420% to −0.077%]).

Most combinations containing a sulfonylurea (without metformin) were associated with reduced A1C values in young-onset T2D (−0.08% [−0.099% to −0.065%] for sulfonylurea alone) but increased A1C values in usual-onset T2D (+0.06% [+0.059% to +0.072%] for sulfonylurea alone). The combination of a sulfonylurea and an α-glucosidase inhibitor was associated with the largest A1C decrement in young-onset T2D (−0.10% [−0.266% to 0.064%]; usual-onset: +0.25% [+0.196% to +0.312%]). Sensitivity analyses excluding A1C values from the first 6 and 12 months after diagnosis yielded relatively similar findings for metformin-based combinations, although combinations containing a sulfonylurea were associated with increased A1C values in young-onset T2D (Fig G in S1 Appendix). Adjustment for the baseline A1C with each OGLD combination yielded similar results to the main analysis (Fig H in S1 Appendix).

## Discussion

In this large population- and register-based study, we found that people with young-onset T2D had poorly controlled hyperglycemia throughout their life span, resulting in more than triple the cumulative glycemic exposure versus usual-onset T2D. This disparity was driven by rapid glycemic deterioration, which was particularly steep among people diagnosed with T2D before age 30 years. Conversely, we revealed that T2D diagnosis after age 60 years was associated with glycemic improvement—a novel finding that has not been previously reported to our knowledge. In this real-world study, people with young-onset T2D had slightly smaller A1C decrements compared with usual-onset T2D for most combinations of OGLDs including metformin, whereas young-onset T2D was unexpectedly associated with greater responsiveness to sulfonylureas and α-glucosidase inhibitors than usual-onset T2D. Although most OGLD combinations appear to lower A1C by similar decrements across age at diagnosis, the rapidity of glycemic deterioration in young-onset T2D suggests that early combination therapy [33] and aggressive treatment escalation are needed to reduce the massive excess in glycemic exposure that we observed.

### Excess glycemic exposure in young-onset T2D

Our study is the first, to our knowledge, to describe the greater than 3-fold disparity in lifetime glycemic exposure between young- and usual-onset T2D. We estimated that people with young-onset T2D will accumulate nearly 40 A1C-years of glycemic exposure by age 75 years. Considering that every 10 A1C-years predicts a 25% increase in relative risk of MACE [2], this excess glycemic exposure is consistent with the high complication risks of young-onset T2D that we [6,7] and others [8,9] have reported. In the United Kingdom Prospective Diabetes Study [1], early reduction of glycemic exposure in T2D has been proven to prevent complications in future decades. Our real-world findings emphasize the urgent need for a paradigm shift toward early and intensive glycemic management to improve survival in the young-onset T2D population.

## Rapid glycemic deterioration in young-onset T2D

Glycemic deterioration was particularly rapid in young-onset T2D, peaking at 0.09% per year for an age at diagnosis of 20 years. This rapid progression is thought to be caused by declining β-cell function in the setting of peripheral and hepatic insulin resistance [34,35]. Because of genetic factors, impaired β-cell secretion is an especially important driver of T2D in East Asians [36]. Our findings are consistent with the 20% to 35% per year decline in β-cell function reported among children aged 10 to 17 years in the Treatment Options for type 2 Diabetes in Adolescents and Youth study [34]. By contrast, we observed glycemic improvement among adults diagnosed at age 60 years or above. This finding supports the principle of therapy de-intensification among older adults [37], who may be predisposed to developing a milder phenotype of T2D [38,39]. However, previous studies of usual-onset T2D in European populations reported different results. In Scotland, glycemic deterioration rates were 0.09% and 0.20% per year for ages at diagnosis of 70 and 50 years, respectively, according to the Genetics of Diabetes Audit and Research in Tayside Study [13]. In A Diabetes Outcome Progression Trial, the rate of glycemic deterioration in a largely European cohort was 0.07%–0.14% per year [12]. It is unclear why these rates differed from our study. Rates of β-cell decline might be lower among older Chinese people [40] compared with Europeans [41]. Lifestyle might also play a role, as Hong Kong has the world's most physically active population [42,43].

## Differential responses to OGLDs between young- and usual-onset T2D

In this observational study, A1C decrements associated with OGLD combinations were relatively modest compared with clinical trials. Although we adjusted for adherence using time-varying covariates based on prescription dispensing records, this discrepancy may relate to people not consuming their dispensed medications [44,45]. Rather than focusing on the absolute magnitude of A1C decrements, we compared relative changes in A1C across different ages at diagnosis and drug classes to identify any potentially informative response differences in this real-world setting. In both young- and usual-onset T2D, the combination of metformin, a sulfonylurea and a DDP-4 inhibitor was associated with the greatest A1C reduction. These findings are consistent with evidence suggesting that dipeptidyl peptidase-4 inhibitors lower blood glucose especially well among of East Asians and other people with low body mass indices [46,47]. This combination might be particularly effective in the context of β-cell dysfunction in East Asians, especially in the presence of obesity [48]. Thiazolidinediones were associated with relatively large A1C decrements when used in combination with metformin and a sulfonylurea, which is consistent with the findings of the Treatment Options for type 2 Diabetes in Adolescents and Youth study [34].

Overall, young-onset T2D was associated with slightly smaller A1C decrements versus usual-onset T2D for most—but not all—regimens. As observed in previous studies [12], combinations containing a sulfonylurea were associated with increased A1C values in usual-onset T2D (0.06%–0.25%). However, in young-onset T2D, sulfonylureas were associated with reduced A1C values overall (−0.08% to −0.10%) and relatively smaller A1C increases compared with usual-onset T2D when excluding the initial 6-month period after diagnosis. The reason for this differential response is unclear. In our setting, we previously reported that 5% of people in a young-onset T2D cohort actually had maturity onset diabetes of the young type 3 [49]—which is known to respond well to sulfonylureas [50]—although genetic data were unavailable in the present study. Additionally, α-glucosidase inhibitors, which reduce glycemic excursion and demand on β-cells, were associated with greater A1C reductions in young- versus usual-onset T2D, albeit with wide confidence intervals. This class is efficacious and well

tolerated in East Asians [51–54] and merits further investigation among people with young-onset T2D and poor β-cell reserve.

## Strengths and limitations

Strengths of our study include its large sample size, complementary cohorts, and long follow-up duration. Unlike previous studies, our detailed register enabled us to capture A1C levels across the lifetime and quantify glycemic exposure at various ages at diagnosis. Using a territory-wide laboratory database containing over 3 million A1C measurements and 2.8 million dispensing records, we were able to demonstrate the important associations between age at diagnosis and glycemic deterioration and responses to OGLDs. Future studies are required to determine whether these findings extend to sodium-glucose cotransporter-2 inhibitors and glucagon-like peptide 1 receptor agonists, which were not widely available in our setting during the study period. As this was a real-world study, we could not confirm whether people consumed the medications dispensed. We cannot exclude the possibility of confounding by indication or by unmeasured variables. Further research is needed to confirm whether these findings are generalizable outside of Hong Kong's healthcare system and to other ethnic groups. The differential responses to OGLDs between young- and usual-onset T2D are hypothesis-generating, and experimental studies are needed to confirm these findings.

## Conclusion

Young-onset T2D progresses rapidly and remains poorly controlled throughout the life span, resulting in more than triple the lifetime glycemic exposure versus usual-onset T2D. The larger A1C reductions associated with sulfonylureas and α-glucosidase inhibitors in young-onset T2D emphasize the importance of β-cell dysfunction in this condition and calls for more precise phenotyping to personalize therapy [55–58]. Pending such evidence, our data suggest that most OGLD regimens are associated with lower glucose levels in young-onset T2D. However, better strategies are needed to apply and rapidly escalate these treatments to overcome therapeutic inertia and mitigate the aggressive trajectory of glycemic deterioration in this high-risk population—especially considering the lack of pharmacologic approaches to improve underlying β-cell function [17]. Given the complex biological and behavioral determinants of young-onset T2D, it is especially important to deploy these strategies in the context of integrated team-based care, disease classification, and patient empowerment to achieve better outcomes [59].

## Supporting information

**S1 Appendix.**
(PDF)

**S1 Analysis plan.**
(PDF)

**S1 STROBE checklist. STROBE, Strengthening the Reporting of Observational Studies in Epidemiology.**
(DOCX)

## Acknowledgments

We thank Dr. Allan Detsky for his useful comments on an earlier version of this manuscript and the Hong Kong Hospital Authority for providing the data for this study. Part of this work

was presented at the American Diabetes Association's 79th Scientific Sessions in San Francisco, California, June 7–11, 2019.

## Author Contributions

**Conceptualization:** Calvin Ke, Thérèse A. Stukel, Baiju R. Shah, Ronald C. Ma, Juliana C. N. Chan, Andrea Luk.

**Data curation:** Eric Lau, Wing-Yee So, Alice P. Kong.

**Formal analysis:** Calvin Ke.

**Funding acquisition:** Juliana C. N. Chan, Andrea Luk.

**Investigation:** Calvin Ke, Elaine Chow.

**Methodology:** Calvin Ke, Thérèse A. Stukel, Baiju R. Shah, Eric Lau.

**Project administration:** Wing-Yee So, Juliana C. N. Chan, Andrea Luk.

**Resources:** Eric Lau, Juliana C. N. Chan, Andrea Luk.

**Software:** Eric Lau.

**Supervision:** Thérèse A. Stukel, Baiju R. Shah, Juliana C. N. Chan, Andrea Luk.

**Writing – original draft:** Calvin Ke.

**Writing – review & editing:** Calvin Ke, Thérèse A. Stukel, Baiju R. Shah, Eric Lau, Ronald C. Ma, Wing-Yee So, Alice P. Kong, Elaine Chow, Juliana C. N. Chan, Andrea Luk.

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
