## [Editor Report · Decision Letter 0]

24 Apr 2020

Dear Dr Ke, 

Thank you for submitting your manuscript entitled "Age at Diagnosis, Glycemic Control, and Responses to Oral Glucose-Lowering Drugs in Type 2 Diabetes: A Population-Based Study" for consideration by PLOS Medicine.

Your manuscript has now been evaluated by the PLOS Medicine editorial staff as well as by an academic editor with relevant expertise and I am writing to let you know that we would like to send your submission out for external peer review.

Kind regards,

Artur Arikainen,

Associate Editor

PLOS Medicine

---

## [Decision Letter · Decision Letter 1]

1 Jun 2020

Dear Dr. Ke,

Thank you very much for submitting your manuscript "Age at Diagnosis, Glycemic Control, and Responses to Oral Glucose-Lowering Drugs in Type 2 Diabetes: A Population-Based Study" (PMEDICINE-D-20-01525R1) for consideration at PLOS Medicine. 

[LINK]

In light of these reviews, I am afraid that we will not be able to accept the manuscript for publication in the journal in its current form, but we would like to consider a revised version that addresses the reviewers' and editors' comments. Obviously we cannot make any decision about publication until we have seen the revised manuscript and your response, and we plan to seek re-review by one or more of the reviewers. 

We expect to receive your revised manuscript by Jun 22 2020 11:59PM. Please email us (plosmedicine@plos.org) if you have any questions or concerns.

We look forward to receiving your revised manuscript. 

Sincerely,

Artur Arikainen, 

Associate Editor 

PLOS Medicine

plosmedicine.org

1. Please address all of the reviewers’ comments below.

2. Title: Please describe your study as ‘observational’, and include the cohort location ‘Hong Kong’.

3. Abstract:

a. Please move the limitations to the end of the ‘Methods and findings’ subsection.

b. Please include basic demographic information for the cohort (age, sex etc.).

c. Please quantify the main results with 95% CIs and p values.

4. Please include line numbers in the margin throughout your manuscript.

6. Results: Please quantify all results with 95% CIs and p values.

7. Please avoid using the word “effect” or similar words when describing associative findings from your observational study.

8. Please remove the Data Availability, Funding, Competing Interests, and Author contributions from after your Acknowledgements – these will be taken from the submission form instead.

9. When completing the STROBE checklist, please use section and paragraph numbers, rather than page numbers. Please add the following statement, or similar, to the Methods: "This study is reported as per the Strengthening the Reporting of Observational Studies in Epidemiology (STROBE) guideline (S1 Checklist)."

10. Did your study have a prospective protocol or analysis plan? Please state this (either way) early in the Methods section.

a. If a prospective analysis plan (from your funding proposal, IRB or other ethics committee submission, study protocol, or other planning document written before analyzing the data) was used in designing the study, please include the relevant prospectively written document with your revised manuscript as a Supporting Information file to be published alongside your study, and cite it in the Methods section. A legend for this file should be included at the end of your manuscript. 

b. If no such document exists, please make sure that the Methods section transparently describes when analyses were planned, and when/why any data-driven changes to analyses took place. 

c. In either case, changes in the analysis-- including those made in response to peer review comments-- should be identified as such in the Methods section of the paper, with rationale.

11. Please state whether the patient data were anonymised at the time of access, or whether patients provided written informed consent for the use of their data.

Comments from the reviewers:

Reviewer #1: I confine my remarks to statistical aspects of this paper.

These were very well done and I recommend publication

Peter Flom

Reviewer #2: 

This is a manuscript which describes the relationship between age at diagnosis, glycemic control, and responses to oral glucose-lowering drugs in type 2 diabetes in a population-based study from Hong Kong.

Analyses examining poor outcomes ( diabetes complications) among those with young onset diabetes versus older onset diabetes are tricky because is difficult to understand whether any relationship observed are due to someone particular about young onset diabetes or merely due to duration. This study partly avoids this conundrum and uses lifetime glycaemia as outcome.

The sample used in this study is very large and extracted from the Hong Kong Diabetes Surveillance Database and Hong Kong Diabetes Registry. The size of the sample is a real strength as other studies on young versus older onset type 2 diabetes are based on clinic samples of much smaller size.

Specific comments:

There are several statements in the abstract which do not have a statement of significance.

Figure 2 is difficult to understand. Does the direction and the colour of the arrow mean something? I assume it does but and I can guess what it means, but it should be specified more clearly. Have you thought about plotting trajectories of HbA1c in classes perhaps? I did not find this figure helpful.

Although the work on glucose lowering drugs among these groups is interesting, it seems a bit like a bit of an add on. Are medications free of charge in Hong Kong? Are there any health access issues which could influence these results? Some context should be provided for those who are not familiar with the health system in Hong Kong. When were SGLT2 inhibitor available in Hong Kong?

How was adherence and persistence dealt with?

The % missing on the tables could be removed. It does not really add anything.

The discussion could include some text about the mechanisms explaining why HbA1c deterioration is faster in young onset type 2 versus older onset type 2 diabetes.

Reviewer #3: The effect of age of onset of diabetes on long term outcomes is an interesting topic given that it may require differential management approaches based on individual risk. The rising prevalence of type 2 diabetes in youth is a matter of major concern. The clinical observation that diabetes in youth has a more aggressive phenotype requires better scrutiny especially to differentiate it from older onset type 2 diabetes. This study addresses 3 major aspects of this issue in relation to the glycemic burden, trajectory and response to treatment. The authors have done well in exploring these key issues in tis study.The manuscript is well written and the methods and results are presented with good clarity. The large sample size and the use of information from well established registers is a strength of this study. Discussion is well written and is balanced.

Couple of minor points : 1. As the cohort is mainly east asian, it would be good to add a statement re generalisability of these findings in other populations .

 2. With regards to treatment comparisons, although the focus was on OGLD was insulin used in combination over the duration of follow up in this cohort? If so, it is worth mentioning given the rapid progression to insulin requirement in younger age groups. It is also important to discuss the point that 24% of younger diabetics were requiring insulin within 1 year of index date.

 3. Reference to the effects of TZDs in younger age groups ( found to be effective on this group in combination with SU and Metformin) in the context of findings of the TODAY trial would be good.

Reviewer #4: PMEDICINE-D-20-01525R1

This is a nice well written paper that explores an area for which there are few data. A significant strength is the large study population and that the study employs valuable and well defined cohorts. The study adds to new knowledge by quantifying excess glycaemia exposure for younger onset type 2 diabetes and further examines the trajectory of glycaemic decline and relative efficacy of therapy. Data are real world and observational so that all conclusions should be seen in this context. Nevertheless the data as presented do align with previous studies in adolescent type 2 diabetes as expressed in the landmark TODAY study and the more recent RISE data. I cannot critique the appropriateness of the statistical methods in full. I do have a number of largely minor comments 

1.The authors should comment on the possibility of ascertainment bias, especially the possibility of more severe young diabetes being referred to relevant service and be preferentially represented in the two datasets (especially the HKDR). Also in the HKDSD what is the possibility of those captured by metformin use having pre-diabetes and or PCOD and not diabetes per se. 

2. Re glucose exposure outcome. Re mean observed HbA1c measured at standard intervals and number of observations standardised. How many HbAic observations were taken by age of diagnosis category? That the lifetime cumulative glucose exposure is greater in young onset compared to usual onset is not counter-intuitive given the longer duration of disease. Or was glucose exposure comparisons equated for duration in this study? Could this be clarified in the manuscript? It might be more informative to express this excess equated for duration of observation. Further how many younger onset individuals contributed to the HbAic data at different current ages up to age 75. This would add perspective on the strength of these data.

3. Re Response to OGLD . The delta Hba1c for any treatment is dependent on starting Hba1c. Can this be or was this adjusted for in this analysis. It is a significant effect modifier. Comment on the quality of the dispensing records would be helpful

4. Re the exclusion of patients in the HKDSD Appendix fig 4 . A large number (>20,000) were excluded as they had incomplete HbAic. It would be nice to know that these data were missing at random and would not have biased outcomes in any way. 

5. The data are from a HK Chinese population. Could the authors comment on generalizability 

6. The excess glycaemic deterioration in the young onset could be due to failure of treatment escalation, therapeutic inertia. This is an important point as data from primary care suggest a greater failure to recognise the importance of treatment escalation in youth. Could the authors comment or shed light on this possibility.

[LINK]

---

## [Decision Letter · Decision Letter 2]

14 Jul 2020

Dear Dr. Ke,

Thank you very much for re-submitting your manuscript "Age at Diagnosis, Glycemic Trajectories, and Responses to Oral Glucose-Lowering Drugs in Type 2 Diabetes: A Population-Based Observational Study in Hong Kong" (PMEDICINE-D-20-01525R2) for review by PLOS Medicine.

I have discussed the paper with my colleagues and the academic editor and it was also seen again by 3 reviewers. I am pleased to say that provided the remaining editorial and production issues are dealt with we are planning to accept the paper for publication in the journal.

[LINK]

We look forward to receiving the revised manuscript by Jul 21 2020 11:59PM. 

Sincerely,

Artur Arikainen, 

Associate Editor 

PLOS Medicine

plosmedicine.org

Requests from Editors:

1. Please update the Title to: “Age at Diagnosis, Glycemic Trajectories, and Responses to Oral Glucose-Lowering Drugs in Type 2 Diabetes in Hong Kong: A Population-Based Observational Study”

2. Please update the Short Title to: “Age at Diagnosis and Glycemic Trajectories in Type 2 Diabetes”

3. Abstract:

a. Please add additional limitation(s), eg. generalisability outside Hong Kong, to the ‘Methods and findings’ subsection.

b. Please do not use italics for emphasis, e.g., at line 38.

c. Please quote exact p values or p<0.001 (e.g., at line 49).

d. Line 51: Delete second “versus”.

e. Please provide quantitative data with 95% CIs and p values for each of the following statements:

i. “Age at diagnosis ≥60 years was associated with glycemic improvement.”

ii. “Responses to OGLDs differed by age at diagnosis (p<0.0001).”

iii. “Those with young-onset T2D had smaller A1C decrements for metformin-based combinations versus usual-onset T2D, but greater responses to other combinations containing sulfonylureas.”

f. Line 59: we suggest adapting the text to "In this study, we observed excess ..." or similar, and segueing into "... indicating that improved treatment strategies are needed in this setting."

4. Lines 423-424: Please avoid using the word ‘effectively’, and re-word as follows: “…our data suggest that most OGLDs regimens are associated with lower glucose levels in young-onset T2D.”

5. Discussion:

a. Line 344: Please clarify that similar findings have not been reported “to the authors’ knowledge”.

b. We note that treatment with GLP-1 agonists seemed low in this cohort - please perhaps comment on lack of information about and/or availability of newer drugs, either as a limitation of the study if these have become more widely used post-study, or of the health system if cost is an issue.

6. Re: reference 39 listed as in press, papers cannot be listed in the reference list until they have been accepted for publication or are otherwise publicly accessible (for example, in a preprint archive). The information may be cited in the text as a personal communication with the author if the author provides written permission to be named. Alternatively, please provide a different appropriate reference.

7. In the attached STROBE checklist, please remove page numbers, and refer to individual items by section (e.g., "Methods") and by paragraph number as at present.

----

Comments from Reviewers:

Reviewer #1: I had already accepted the earlier revision, so I recommend publication

Peter Flom

Reviewer #2: Thanks for addressing the comments and simplifying the figure.

Reviewer #3: I have now reviewed the manuscript after the revisions. I can see that the authors have addressed the comments from the reviewers satisfactorily. Figure 1 and Figure 2 are much better now. Discussion is well balanced. I would be delighted to see this work published.

[LINK]

---

## [Editor Report · Decision Letter 3]

14 Aug 2020

Dear Dr. Ke, 

On behalf of my colleagues and the academic editor, Dr. Sanjay Basu, I am delighted to inform you that your manuscript entitled "Age at Diagnosis, Glycemic Trajectories, and Responses to Oral Glucose-Lowering Drugs in Type 2 Diabetes in Hong Kong: A Population-Based Observational Study" (PMEDICINE-D-20-01525R3) has been accepted for publication in PLOS Medicine. 

PRODUCTION PROCESS

PRESS

PROFILE INFORMATION

Thank you again for submitting the manuscript to PLOS Medicine. We look forward to publishing it. 

Best wishes, 

Artur Arikainen, 

Associate Editor 

PLOS Medicine

plosmedicine.org